# Effects of a Physical Literacy Breaks (PLBreaks) Program on Physical Literacy and Body Composition in Portuguese Schoolchildren: A Study Protocol

**DOI:** 10.3390/biology11060910

**Published:** 2022-06-13

**Authors:** Maria Mendoza-Muñoz, Jorge Carlos-Vivas, Santos Villafaina, Jose A. Parraca, Alejandro Vega-Muñoz, Nicolás Contreras-Barraza, Armando Raimundo

**Affiliations:** 1Research Group on Physical and Health Literacy and Health-Related Quality of Life (PHYQOL), Faculty of Sport Sciences, University of Extremadura, 10003 Cáceres, Spain; mamendozam@unex.es; 2Departamento de Desporto e Saúde, Escola de Saúde e Desenvolvimento Humano, Universidade de Évora, 7004-516 Évora, Portugal; jparraca@uevora.pt (J.A.P.); ammr@uevora.pt (A.R.); 3Promoting a Healthy Society Research Group (PHeSO), Faculty of Sport Sciences, University of Extremadura, 10003 Cáceres, Spain; 4Physical Activity and Quality of Life Research Group (AFYCAV), Faculty of Sport Science, University of Extremadura, 10003 Cáceres, Spain; 5Comprehensive Health Research Centre (CHRC), University of Évora, 7004-516 Évora, Portugal; 6Public Policy Observatory, Universidad Autónoma de Chile, Santiago 7500912, Chile; alejandro.vega@uautonoma.cl; 7Facultad de Economía y Negocios, Universidad Andres Bello, Viña del Mar 2531015, Chile; nicolas.contreras@unab.cl

**Keywords:** active recess, children, daily activity behaviour, knowledge, motivation, physical competence, students

## Abstract

**Simple Summary:**

Active breaks have led to improvements in physical fitness, daily steps taken or even improvements in the health of participants who have taken them. However, no study has assessed how they affect physical literacy (PL). This study will try to explore the effect of a programme based on active breaks (PLBreaks) on PL and body composition of schoolchildren. For this purpose, the PLBreaks programme will be carried out for 3 months and 3 days a week. The PLBreaks programme will consist of two blocks of 10 min of different physical activities (PA). The first block will be focused on the acquisition of knowledge and healthy life habits that will contribute to the development of the domains of knowledge and understanding and daily activities. The second block will be focused on the physical competence and motivation through games. Furthermore, a control group who will follow their usual daily activities is included. This would allow the investigation of whether the PLBreaks programme leads to an improvement in PL and body composition in schoolchildren. If the effectiveness of the programme is demonstrated, the programme could be included in public education programmes, representing a scientific advance in terms of improving health-related PA and adherence, as well as the prevention of diseases associated with inactivity.

**Abstract:**

(1) Background: Several studies have shown that active breaks have led to different improvements in their participants. However, no studies have assessed how they affect physical literacy (PL). (2) Aims: Therefore, this study will examine the effect of the PLBreaks programme on school children’s PL and body composition. (3) Methods: A parallel-group randomised controlled trial will be conducted with assessments of PL (Canadian Assessment of Physical Literacy Development) and body composition (height, bodyweight, fat mass and fat-free mass) before and after an active breaks programme. PLBreak programme will run for 3 months and will be carried out 3 days a week for 20 min each day. The PLBreaks programme will consist of two blocks of 10 min of different physical activities (PA). The first block will be focused on the acquisition of knowledge and healthy life habits that will contribute to the development of the domains of knowledge and understanding and daily activity. The second block will be focused on physical competence and motivation throughout games. (4) Conclusions: The present study will investigate the efficacy of PLBreaks in schoolchildren in improving their PL and body composition. If the efficacy of the program is demonstrated, including the programme in public education programmes can be possible. This could be a scientific breakthrough in terms of health-related PA improvement and adherence, as well as the prevention of diseases associated with inactivity.

## 1. Introduction

The World Health Organisation (WHO) defines overweight and obesity as chronic diseases characterised by abnormal or an excessive accumulation of body fat that can be harmful and represent a health risk. Childhood obesity has increased in the last 40 years, making it one of the major problems and challenges for public health in advanced societies [1,2,3]. Focusing on Portuguese population, previous studies showed that the prevalence of overweight in children and adolescents ranges between 20% and 40%, and obesity between 10% and 15% [4]. Likewise, the Childhood Obesity Surveillance Initiative (COSI) report for Portugal in 2019 showed a reverse trend in the prevalence of overweight (from 37.9% to 29.6%) and obesity (from 15.3% to 12.0%) in children aged 6 to 8 years-old [5]. However, data are still worrying, as the National Food, Nutrition and Physical Activity survey 2015–2016 (IAN- AF, 2015–2016) informed that the prevalence of overweight and obesity was 25% in children, and even higher in adolescents, reaching 32.3% [6]. 

Multiple factors influence the development of overweight and obesity, such as genetic, neuroendocrine, energy expenditure-related and environmental factors [7]. Specifically, one of the major problems in current society is the high level of sedentary lifestyles, which is considered the disease of the 21st century [8,9]. In Portugal, IAN-AF, 2015–2016 [6] published that only the 57.5% of children and adolescents (aged 6–14 years) met WHO recommendations (60 min/day of moderate/vigorous physical activity; i.e., ≥3 METS/hour), and this compliance is higher in children versus adolescents (68.3% versus 57.1%). 

Hence, due to the numerous problems that physical inactivity can lead to and the risks it poses to health in both the short and long terms, it is a challenging problem for society. The health benefits of physical activity (PA) and its importance in disease prevention have already been demonstrated [10]. Moreover, it is estimated that physical fitness is one of the most relevant markers of health [11] as well as a significant predictor of cardiovascular disease mortality and morbidity [12,13,14,15]. Although genetic inheritance determines part of the individual’s physical fitness, it is also influenced by environmental factors and lifestyles. This is where physical exercise became relevant [11], as numerous studies have reported benefits in terms of body composition after a PA intervention [16,17]; thus, prevention at this age could be essential for avoiding an increase in health risks.

Physical literacy (PL) can play a relevant role in enhancing physical fitness since it can influence the practice of physical activity. PL was defined in the Bulletin of the International Council of Sport Science and Physical Education of the United Nations Educational, Scientific and Cultural Organization as the motivation, confidence, physical competence, knowledge and understanding to value and participate in a physically active lifestyle [18]. Similarly to how reading, writing, listening and speaking are combined to formulate linguistic literacy, PL is a progressive journey in which the different components interact holistically to facilitate a life of participation and enjoyment of PA.

The positioning of PL as a determinant of health is not a novel idea, but little focus has been placed on what it means and its implications. Few studies have investigated the association between PL and health, but it has already been shown to be related to body composition [19], physical fitness, blood pressure and HRQoL [20]. Active breaks may be highly relevant for the acquisition of PA benefits. Previous studies have shown that active breaks based on educational and health interventions have led to improvements in participants’ fitness and health [21,22], self-efficacy [23], quality of life and self-confidence [24].

In Portugal, only one study [25] related to PL has been found, which aimed to validate a battery for its evaluation. However, this battery was focused on adolescents aged between 15 and 18 years; thus, there are currently no studies that evaluated PL in children and no studies that have developed programmes aimed at improving PL. Despite this, we consider this multidimensional assessment and intervention model to be of great interest. 

Thus, this study aimed to (1) evaluate the effectiveness of the Physical Literacy Breaks (PLBreaks) programme on schoolchildren’s PL (general PA, physical competence, motivation and knowledge domains) and (2) evaluate the effectiveness of the PLBreaks on children’s body composition.

## 2. Materials and Methods

### 2.1. Design

A parallel-group randomised controlled trial will be conducted including a 3-month intervention phase, followed by 1-month follow-up. Participants will be randomly assigned to the experimental or control group. For both groups (control and experimental), assessments will be carried out at baseline (before starting the intervention), right after finishing the intervention and an additional follow-up assessment 1 month after the end of the programme. The study will be conducted following the *Consolidated Standards of Reporting Trials Statement* (CONSORT) [26].

### 2.2. Ethics

The Ethics Committee of the University of Evora approved this project (approval number: 22047) according to the updates of the Declaration of Helsinki and modified by the 64th General Assembly of the World Medical Association (Fortaleza, Brazil, 2013) and the Law 14/2007 on Biomedical Research. The study has been registered in the Australian New Zealand Clinical Trials Registry (Registration Number: ACTRN12622000562774; https://www.anzctr.org.au/ (accessed on 22 May 2022)).

### 2.3. Sample Size

Sample size computations were conducted by using G*Power software 3.1.9.4 (Kiel University, Kiel, Germany) and selecting the statistical test to compare the difference between two independent means (two groups). Thus, accepting a 0.05 alpha risk and a 0.2 beta risk in a bilateral contrast and assuming a moderate effect size of 0.5 [27], a total of 128 participants (64 subjects in the experimental group and 64 in the control group) were sufficient for achieving a minimum power of 80%. 

### 2.4. Randomisation and Blinding

Participants will be randomly assigned to the experimental (PLBreaks program) or control groups. Prior to enrolling participants (1:1), the Research Randomizer software (version 4.0, Geoffrey C. Urbaniak and Scott Plous, Middletown, CT, USA; http://www.randomizer.org) [28] will be employed to create a randomisation sequence. A member of the research team, with no active clinical involvement in the trial, will conduct this process. Group assignment will be hidden in a password-protected computer file. Participants will be aware of their group assignment, but outcome assessors and data analysts will be blinded to the participants group allocation.

### 2.5. Participants

Participants have to meet the following inclusion criteria: (1) to be aged between 8 and 16 years old; (2) to be registered and/or residing in Portugal; (3) not suffer from pathologies that contraindicate exercise practice, limit the execution of the PA programme or require special attention; (4) to be authorised by parents or legal guardians; and (5) to participate in the study.

### 2.6. Intervention

Experimental group: A 3-month physical active breaks intervention will be carried out in the educational centres during break times, with a frequency of three times per week. The duration of each session will be between 20 min, based on the time pre-established by each educational centre for the students’ break.

Specifically, in order to develop the domains that make up physical literacy, the PLBreaks programme will consist of two blocks of 10 min. Both blocks will favour the development of physical competence through different physical activities. In addition, the first block will be focused on the acquisition of knowledge and healthy life habits that will contribute to the development of the domains of knowledge and understanding and daily activity. Finally, the second block, in addition to favouring physical competence, will try to improve motivation through different strategies that will be detailed below. 

In the first block, activities will be carried out to develop the knowledge and understanding of concepts and attitudes related to physical activity (Figure 1). This will be performed with the following activities:Find the question: Participants, in pairs, will have a map with different points marked on it (beacons). They will have to find the maximum number of beacons and answer the questions that are exposed in these points on a record sheet.Body performances: Participants in groups will have to act out the content of the week through body movements and their partners will have to guess their actions. Card relay race: Participants will be divided into teams, which will be placed in rows. Each participant of each team will have to run individually to a table located about 30 metres away, and once there, they will find a series of cards, and will have to place the card they will be holding on their corresponding (partner) and return to the end of the line for the next participant to come out. 

The second block will be based on active play, as it is fundamental in the transmission of values and attitudinal content as well as stimulating social and civic relations with others [29]. Similarly to previous studies [30,31], for this study, the proposed Ontario Physical and Health Education Association’s PlaySport activities (http://www.playsport.net/ (accessed on 20 May 2022)) will be used (Figure 1). During the games, basic skills will be developed, such as moving, jumping, spinning or throwing, thus trying to improve them as well as the children’s motivation. The games will have a cooperative and competitive component due to the motivational benefits this can bring [32]. The activities will be adapted to the initial level of the children’s.

Specifically, during the second block, in order to improve the motivation and confidence of the participants, some strategies should be fulfilled. In this regard, participants’ inclusion will be ensured during all activities (both by gender and ability) by adapting all activities so that all participants can learn and participate actively. Furthermore, to satisfy the need for peer relationships, strategies will be used in group formation and cooperative activities with a common goal that participants will have to achieve together. [33]. In addition, activity coordinators and children will be encouraged to provide verbal feedback to participants in order to ensure active participations, fostering relation-inferred self-efficacy (RISE) [30,34]. Finally, they will be provided time at the end of each session to express their opinions [33].

**Figure 1 biology-11-00910-f001:**
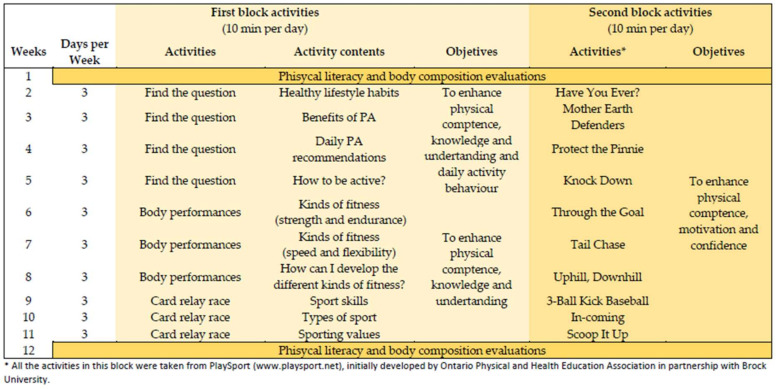
Activities and contents to be developed during the PLBreaks programme.

Control group: Participants will continue with their normal daily routine during school breaks (interacting among peers and playing freely or eating their snacks), which will not include any physical exercise similar to the exercise practiced by the experimental group.

### 2.7. Measures

A variety of tools will be used to assess the PL and body composition of schoolchildren. Before the first measurement, all participants will undergo a familiarisation phase to become acquainted with the different instruments and assessments included in this project.

a.Anthropometric and body composition measurements:

Height, bodyweight, fat mass and fat-free mass will be taken at baseline at the end of the intervention (month 4) and one month after the end of the intervention (follow-up).

Anthropometric measurements will be taken under standardised conditions and following WHO recommendations for the development of the Childhood Obesity Surveillance Initiative (COSI) [35]. Before starting any of the measurements, participants will be asked to remove their shoes and socks, as well as any heavy clothing (coats, jumpers, jackets, etc.). They will also be asked to empty their pockets and remove their belts and any other accessory (headbands, pendants, etc.). 

Height shall be measured with a measuring rod (Tanita Tantois, Tanita Corporation, Tokyo, Japan). The instrument shall be placed on a vertical surface with the measuring scale perpendicular to the ground. Participants will be positioned on a standing position, with the shoulders balanced and the arms relaxed along their body. The head will be held in the Frankfurt plane. Height will be taken in cm and rounded to the nearest mm. 

Bodyweight will be measured using a bioimpedancemeter (Tanita MC-780 MA, Tanita Corporation, Tokyo, Japan). The assessment shall be performed in ‘standard mode’ by entering the participant’s age, sex and height. Bodyweight will be recorded in kg, with an accuracy of 100 g. This instrument will also be used to assess participants’ fat mass, fat mass percentage and fat-free mass. To obtain accurate measurements from the bioimpedancemeter, preconditions will be standardised according to the manufacturer’s guidelines [36,37]: (a) more than 3 h after waking up; (b) urinating before the measurement; (c) not eating or drinking within the last 3 h; (d) not eating or drinking excessively the previous day; (e) no vigorous exercise within the last 12 h; (f) no alcohol consumption within the last 12 h; and (g) no use of metal objects or wearing a pacemaker.

BMI will be determined using the following formula: weight × height^−2^. The criteria of the COSI initiative [35] and WHO growth standards [38] will be used to establish weight categories. Participants’ weight status will be established based on the standard deviation (SD) criteria: (1) <−3 SD, “underweight”; (2) <−2 SD, “normal weight”; (3) >+1 SD, “overweight”; and (4) >+2 SD, “obese”. 

b.Physical Literacy:

The level of PL will be assessed at the beginning, at the end of the intervention and one month after the end of the programme by applying the Canadian Assessment of Physical Literacy Development (CAPL-2) [39]. This assessment includes 4 domains: (1) daily physical activity behaviour, (2) physical competence, (3) motivation and confidence and (4) knowledge and understanding. Each domain will consist of a score and different tests. 

Every domain comprises CAPL-2 and the methods detailing how they will be scored are described below.

Daily activity behaviour. The total score will be calculated from two components: step counts obtained using an activity wristband (Xiaomi mi Band 3, Xiaomi Corporation, Beijing, China). These wristbands will record steps for a full week, and a self-reported question on the number of days active for at least 60 min. The total score for this domain will be composed of the score based on the number of steps recorded and the score assigned to the response to the self-reported question on the number of minutes of weekly activity.

Physical competence. The final score for this domain will be reached by the sum of the scores of three components: -*Plank* [40], which consists of maintaining the plank position for as long as possible;-*Progressive Aerobic Cardiovascular Endurance Run* (PACER) test [41], which will allow the measure of cardiorespiratory competence, using a 20 m running test (out and back), following an acoustic signal that determined the intensity of each test stage;-*Canadian Agility and Movement Skill Assessment* (CAMSA) [42], which will test the motor skills of participants through an agility circuit, including throwing, jumping and moving actions.

All tests will be evaluated with a possible score from 1 to 10 points. Thus, a total score of 30 points can be obtained for this domain.

Motivation and confidence. It will be assessed by using the CAPL-2 motivation and confidence questionnaire [39]. It will assess participants’ confidence in the ability to be physically active and motivation to participate in PA. The score will be obtained by the sum of four different dimensions assessed from 1 to 7.5 points: intrinsic motivation, competition, predilection and adequacy. The total score for the domain will be between 1 and 30 points. 

Knowledge and understanding. This domain will assess knowledge about PA [43]. Its score will be obtained from the questionnaire included in the CAPL-2 manual, which includes five questions, four of which are multiple-choice questions and will be scored from 0 to 1. The last question will consist of filling in missing gaps to complete a story and it will be scored from 1 to 6.

Finally, the numerical scores for CAPL-2 will be between 0 and 100 points. Based on these scores, participants will be classified into four levels by considering their sex and age: “insufficient”, “in progress”, “sufficient” and “excellent”. The “insufficient” and “in progress” levels will correspond to children who have not yet reached the optimal level of PL. The “sufficient” level will correspond to children who have reached a score associated with sufficient PL. Finally, the “excellent” level will demonstrate a high level of PL.

### 2.8. Statistical Analysis

Descriptive statistics and computations will be performed with SPSS (version 25.0; IBM SPSS Inc., Armonk, IL, USA). Data will be presented as means and standard deviation (SD).

The normality and homogeneity of data will be checked applying Kolmogorov–Smirnov and Levene’s test, respectively. Then, a repeated measures ANCOVA will be applied to analyse the intervention effects on the different dependent variables, adjusted by age and baseline outcomes. Cohen’s d (with 95% confidence interval) will be also included in the results as the effect size. Effect sizes thresholds will be interpreted as follow: >0.2, small; >0.5, moderate; >0.8, large [27]. Statistical significance will be computed for the effect of time and the interaction group × time. Alpha level will be fixed at *p* ≤ 0.05.

## 3. Discussion

A great interest in the concept of PL is emerging due to its comprehensive character on the development of physical activity. Several studies have started to evaluate and design intervention programmes based on it [30,31,44,45]. However, most of them are focused on some of the domains that comprised PL and not overall. Moreover, most programmes have been carried out within physical education classes [44,45] or in out-of-school settings [30,31]. This project would be the first to implement a programme to develop PL during the break period of the school day and, to the best of our knowledge, the first in Portugal to assess PL in children. 

In Portugal, around 35% of Portuguese people over 14 years old never or rarely take active breaks (e.g., walking, standing or jogging) during sitting time [6]. Break times may be an ideal time for the development of PL, as it has been shown that education and health interventions can improve the physical fitness and health of their participants [22]. In addition, this intervention would contribute towards meeting the World Health Organisation’s physical activity recommendations, which state that children aged 5–17 years should engage in at least 60 min of Moderate to Vigorous Physical Activity (MVPA) per day [46]. It has also been shown that the daily steps taken by students participating in an active school break can be increased compared to those who do not [47,48]. This would help fulfil the daily recommendations set out by the President’s Council on Physical Fitness and Sports, recommending 12,000 steps for girls and 15,000 steps for boys [49]. 

All these benefits have recently led to the development of numerous research studies, which implemented active breaks programmes with different objectives based on different activities, such as alternative, popular and traditional games [47]; free play [50]; pre-sports games [51]; and high-intensity interval training (HIIT) [52] o HIIT + intermittent training [53]. However, any previous study has aimed to develop PL into the school context.

Regarding level of PL, several studies have provided results in schoolchildren from different countries [54,55,56,57,58] based on CAPL-2. Most reported worrying results since more than 50% of the schoolchildren participating in these studies did not achieve sufficient PL [54,55,56,58]. 

More specifically, concerning the “daily activity” domain, which is one of the most relevant due to the consideration of sedentary behaviour as a disease of the 21st century [8,9], previous studies using CAPL-2, observed that the majority of schoolchildren (more than 50%) are in the “insufficient” or “in progress” categories [54,55,56,57]. These results are in line with IAN-AF 2015–2016 outcomes [6], which stated that children and adolescents aged 6–14 years spend on average 9 h and 6 min daily in sedentary behaviours (excluding sleep time) [6]. Moreover, the average time spent in these behaviours increases significantly with age, proceeding from around 8 h in the youngest group and close to 10 h in adolescents [6]. This highlights the importance of promoting active habits at school age, as physical inactivity during the first years of life is considered a relevant factor in the increase in obesity levels and other serious medical disorders in children and adolescents [59].

A previous study has shown that “motivation and confidence” and “physical competence” domains are the CAPL domains with the strongest correlation [60]. Specifically, recent studies found that the greater the percentage of schoolchildren in the domain “motivation and confidence”, the greater the percentage for “physical competence” [57,58]. Other studies have shown that physically active students scored higher on autonomous motivation than non-active students [61,62] and that students who satisfied the needs for autonomy, competence and relatedness were more likely to participate in PE-related activities for self-determined reasons and were less likely to feel demotivated [61]. This highlights the role that motivation plays in the practice and possible adherence to physical activity and, hence, the importance of its assessment in being able to detect possible deficits and, as a consequence, redirect intervention strategies.

Regarding the “knowledge and understanding” domain, previous studies showed that most of the schoolchildren were “in progress” or were “insufficient” [55,56,63]. In this sense, there are hardly any studies that evaluate how knowledge of PA affects subsequent practice. In contrast, other studies that have evaluated how knowledge of certain pathologies affects subsequent treatment have obtained beneficial results [64,65]. Therefore, due to the worrying results reported by the different studies on low knowledge and understanding in this field [54,55,56,58], this study could be very beneficial as an increase in the knowledge about PA could translate into an increase in PL since a relationship has also been found between the two [60]. 

Body composition can have a considerable influence on PL, as several studies have reported significant differences between overweight and non-overweight schoolchildren for the different domains that make up PL [66,67,68,69,70,71,72]. Specifically, in the “daily activity” domain, it has been observed that schoolchildren with obesity performed less daily activity related to physical activity than schoolchildren without obesity [66]. For “physical competence”, it has been shown that overweight schoolchildren have worsened physical competence compared to their normal-weight counterparts [67,68,69,70,71]. In the same line, motivation towards physical activity has been found to be lower in overweight schoolchildren [72]. Regarding the level of knowledge about physical activity, few studies have evaluated this factor in isolation among overweight and obese schoolchildren. However, Nyström, Traversy, Barnes, Chaput, Longmuir and Tremblay [60] reported differences, showing higher knowledge in non-overweight boys. 

Regarding PL and body composition, previous studies have reported a negative correlation between PL and BMI, fat mass [20,58], HRQoL and blood pressure [20]. Nyström, Traversy, Barnes, Chaput, Longmuir and Tremblay [60] and Mendoza-Muñoz, Barrios-Fernández, Adsuar, Pastor-Cisneros, Risco-Gil, García-Gordillo and Carlos-Vivas [58] found that children with a healthy weight scored significantly better on the CAPL than overweight children. Therefore, it can be hypothesised that the development of PL could translate into improved body composition values.

Therefore, the development of educational programmes that promote both knowledges in the field of physical education and the promotion of active and healthy lifestyle habits, as well as considering motivation in the development of these habits, could favour a complete development of PL. This is not limited to the mere development of physical fitness but also of individual fitness. The positioning of PL as a determinant of health is not a novel idea, but little attention has been paid to what this means and its implications [73]. In Portugal, there are no studies in which PL in children is being assessed nor programmes that aim to develop it. However, we consider this model of assessment and multidimensional work to be of great interest, so this study aims to be a starting point to stimulate empirical research on PL. This intervention program will allow children to participate in structured and full physical activity. Interestingly, given that the emphasis of PL is on maximising individual attributes, everyone can achieve it regardless of the initial skill set or level of physical fitness [74]. 

In this sense, the intention of PL proceeds beyond simply being able to define or explain the concept of healthy living but also involves enabling an individual to actualise their inherent potential to adopt and develop a healthy lifestyle [75]. The entrenchment of PL in the educational system would allow the consolidation of a unifying term (and similar to other educational subjects) to describe the overall outcome of quality physical education, physical activity, sport and recreation programmes. Therefore, PL could provide a basis for sport, public health and physical education, which involves trying to promote PL in all areas and addressing it from the early childhood stage so that physical activity can be enjoyed and adhered to from an early age.

One of the main strengths of this study is that the assessment uses CAPL. This instrument is one technique that is most closely aligned to the concept of a child’s PL, as it assesses daily activity, motivation and confidence, knowledge and understanding and physical competence. In addition, it has been shown to be a valid and reliable instrument in school children and, recently, in adolescents. Furthermore, in the last year, it has started to be used in countries such as Denmark [63], Greece [55] and China [56], which will allow for future comparisons between different countries.

The present study does not report preliminary data; this study only reports the study protocol that will be performed in the future. This can be considered as a limitation of the study. As future research lines, we propose the evaluation of PL in adolescent individuals and monitoring its evolution from childhood to this stage. Moreover, it will also be interesting to analyse the differences in the level of PL between stages and to identify relationships between PL domains and health-related outcomes. This would allow the detection of deficiencies and guide interventions to prevent diseases such as obesity.

## 4. Conclusions

The present study will investigate the efficacy of PLBreaks in schoolchildren to improve their PL and body composition. If the efficacy of the program is demonstrated, it can be implemented in schools.

Therefore, the main added value of this study is the acquisition of new knowledge concerning PL from an immediate perspective of practical and direct application. It will represent a scientific advance in terms of improvement and adherence to health-related PA, as well as the prevention of diseases associated with physical inactivity such as overweight and obesity.

On the other hand, the PLBreaks could be the response for solutions to a social problem such as the difficulties of access in many cases (due to socio-economic reasons) to PA programmes that are related to health. Thus, given that it is a low-cost programme that can be easily standardised by levels of difficulty (adapting to the different levels of PA), its transfer and implementation would not be a problem in the public sphere, and it includes the possibility of including the intervention in public education programmes.

## Data Availability

Not applicable.

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
