# Peer review of "Effects of a Physical Literacy Breaks (PLBreaks) Program on Physical Literacy and Body Composition in Portuguese Schoolchildren: A Study Protocol"

_biology, 2022, doi:10.3390/biology11060910_

Round 1
Reviewer 1 Report
Thank you very much for the opportunity to see the work, it is extremely interesting and takes up a very important element of prevention. However, the authors could supplement the literature with the results of research carried out in other European countries.
I suggest supplementing the literature with the following items:
1. Rutkowski T, Sobiech K, Chwałczyńska A. The effect of 10 weeks of karate training on the weight body composition and FFF index of children at the early school age with normal weight and overweight. Arch Budo 2020; 16: 211-219
1. Chwałczyńska A, Rutkowski T, Jędrzejewski G, Wójtowicz D, Sobiech KA. The comparison of the body composition of children at the early school age from urban and rural area in southwestern Poland. BioMed Res Int 2018, Article ID 9694615, 9 s. DOI: 10.1155/2018/969461
1. Chwałczyńska A, Jędrzejewski G, Sobiech KA. The Influence of a Therapeutic Programme on the Segmentary Body Composition in over-and Underweight Children at the Early-School Age: Pilot Studies. J Child Adolesc Behav 2017;5(5):359
Author Response
Dear Reviewer,
Thank you for your review of our manuscript. We have carefully considered your comments and believe that the quality of the paper has improved after incorporating your suggestions. Below are our responses to your suggestions:
“I suggest supplementing the literature with the following items:
- Rutkowski T, Sobiech K, Chwałczyńska A.The effect of 10 weeks of karate training on the weight body composition and FFF index of children at the early school age with normal weight and overweight. Arch Budo 2020; 16: 211-219
- Chwałczyńska A,Rutkowski T, Jędrzejewski G, Wójtowicz D, Sobiech KA. The comparison of the body composition of children at the early school age from urban and rural area in southwestern Poland. BioMed Res Int 2018, Article ID 9694615, 9 s. DOI: 10.1155/2018/969461
- Chwałczyńska A, Jędrzejewski G, Sobiech KA. The Influence of a Therapeutic Programme on the Segmentary Body Composition in over-and Underweight Children at the Early-School Age: Pilot Studies. J Child Adolesc Behav 2017;5(5):359”
- Author’s response: we have included most of them, as suggested.
Reviewer 2 Report
Authors:
Thank you for the opportunity to review the article” Effects of a Physical Literacy Breaks (PLBreaks) program on 2 physical literacy and body composition in Portuguese school-3 children. A study protocol”
Overall, it is a good manuscript, however I just find some specific typographic errors.
2) Introduction
- It is a good introduction. The introduction briefly describes the study and purpose. It also includes the current state of the research field providing sufficient background, and the main aim of the work.
- Just one typographic error in line 98, I supposed you mean benefits
3) Methods
- Another one typographic error in line 166, cooperative
Author Response
Dear Reviewer,
Thank you for your review of our manuscript. We have carefully considered your comments. Below are our responses to your suggestions:
“Overall, it is a good manuscript, however I just find some specific typographic errors.
2) Introduction:
It is a good introduction. The introduction briefly describes the study and purpose. It also includes the current state of the research field providing sufficient background, and the main aim of the work.
- Just one typographic error in line 98, I supposed you mean benefits
3) Methods: Another one typographic error in line 166, cooperative”
- Author’s response: Modified, as suggested.
Reviewer 3 Report
Lines 30-31: '' It will also compare the results of the participants in the program with a control group who will just go about their daily activities '' .... my question is what do the authors mean when they say that the control group carries out daily activities? Please list these types of activities for a clearer understanding of their content.
I think we all know that active breaks are most indicated after exercises / physical activities, etc., especially since they maintain a high level of excitability of the central nervous system. I don't think we need to repeat the importance and the role of active breaks every time. The authors have been repeating their meaning for too many hours, see Simple Summary and Abstract.
''A parallel-group randomised controlled trial will be conducted with assessments of PL (Canadian Assessment of Physical Literacy Development) and body composition (height, body- weight, fat mass and fat-free mass) before and after an active breaks programme''...what would be happen if we gave passive breaks? can PL still be evaluated? If so, under what conditions? if not, what would be the reason? please be detailed
References it may pass in square brackets, not in round brackets, please correct throughout the article.
''Control group: Participants will continue with their usual recreational activities during school breaks''....ok, but please list what they are, it is very important for the readers to understand all these things.
The subchapters ''Intervention'' and ''Measures'' are well presented, for the readers' understanding.
Author Response
Dear Reviewer,
Thank you for your review of our manuscript. We have carefully considered your comments and believe that the quality of the paper has improved after incorporating your suggestions. Below are our responses to your suggestions:
Lines 30-31: '' It will also compare the results of the participants in the program with a control group who will just go about their daily activities '' .... my question is what do the authors mean when they say that the control group carries out daily activities? Please list these types of activities for a clearer understanding of their content.
- Author’s response: Based on your comments, we have clarified this issue in the "intervention" section, in the following answers we specify the changes.
I think we all know that active breaks are most indicated after exercises / physical activities, etc., especially since they maintain a high level of excitability of the central nervous system. I don't think we need to repeat the importance and the role of active breaks every time. The authors have been repeating their meaning for too many hours, see Simple Summary and Abstract.
- Author’s response: We have simplified this as suggested
''A parallel-group randomised controlled trial will be conducted with assessments of PL (Canadian Assessment of Physical Literacy Development) and body composition (height, body- weight, fat mass and fat-free mass) before and after an active breaks programme''...what would be happen if we gave passive breaks? can PL still be evaluated? If so, under what conditions? if not, what would be the reason? please be detailed
- Author’s response: Thank you for your comment. During our study, the control group will take passive breaks, i.e. they will continue with their usual activity during school breaks. This control group will have their PL assessed before and after the same period of time as the intervention in the experimental group. Thus, we will be able to compare between those who did the intervention and those who did not follow any programme, i.e. continued with their usual break activities. Therefore, we have modified the design to clarify this aspect. If you think that further specification is needed, please let us know.
References it may pass in square brackets, not in round brackets, please correct throughout the article.
- Author’s response: Modified, as suggested.
''Control group: Participants will continue with their usual recreational activities during school breaks''....ok, but please list what they are, it is very important for the readers to understand all these things.
- Author’s response: thanks for your suggestion. We have added some examples and also modified the text. We have replaced " usual recreational activities " with " normal daily routine: “Participants will continue with their normal daily routine during school breaks (interacting among peers, playing freely or eating their snacks), which will not include any physical exercise similar to the exercise practised by the experimental group”
- A) Pedro Ángel, L. R., Beatriz, B. A., Jerónimo, A. V., & Antonio, P. V. (2021). Effects of a 10-week active recess program in school setting on physical fitness, school aptitudes, creativity and cognitive flexibility in elementary school children. A randomised-controlled trial. Journal of Sports Sciences, 39(11), 1277-1286.
- B) Drummy, C., Murtagh, E. M., McKee, D. P., Breslin, G., Davison, G. W., & Murphy, M. H. (2016). The effect of a classroom activity break on physical activity levels and adiposity in primary school children. Journal of paediatrics and child health, 52(7), 745-749.
- C) Özdemir, M., Ilkım, M., & Tanır, H. (2018). The Effect of Physical Activity On Social Adaptation And Skills Development In Mentally Disabled Individuals. European Journal of Physical Education and Sport Science.
If you think that more specificity is needed, please let us know.
The subchapters ''Intervention'' and ''Measures'' are well presented, for the readers' understanding.
Reviewer 4 Report
Even if the topic is interesting and could provide some new information in the literature the appropriateness of the study design and the description of the intervention are very deficient. The protocol, which is the key part of the study, is too general.
After a careful consideration, I believe that in this form the manuscript is not suitable for publication in this journal.
Simple summary. Lines 27-29 are very confused.
Abstract:
line 39 (2) Aims: please delete ‘Therefore’
Lines 45-46. As for the simple summary, this part is confused and need to be better clarified: How motivation will be fostered? and how knowledge about healthy lifestyle habits and PA concepts and its benefits will be acquired?
Introduction
Line 61. More recent references are needed.
See for example:
Worldwide trends in body-mass index, underweight, overweight, and obesity from 1975 to 2016: a pooled analysis of 2416 population-based measurement studies in 128·9 million children, adolescents, and adults.
NCD Risk Factor Collaboration (NCD-RisC).Lancet. 2017 Dec 16;390(10113):2627-2642. doi: 10.1016/S0140-6736(17)32129-3. Epub 2017 Oct 10.
Heterogeneous contributions of change in population distribution of body mass index to change in obesity and underweight. NCD Risk Factor Collaboration (NCD-RisC).Elife. 2021 Mar 9;10:e60060. doi: 10.7554/eLife.60060.
Line 100. One of the references is quite old. More recent references are needed.
See for example:
Understanding the Benefits of Brief Classroom-Based Physical Activity Interventions on Primary School-Aged Children's Enjoyment and Subjective Wellbeing: A Systematic Review.
Papadopoulos N, Mantilla A, Bussey K, Emonson C, Olive L, McGillivray J, Pesce C, Lewis S, Rinehart N.J Sch Health. 2022 May 23. doi: 10.1111/josh.13196. Online ahead of print.
Classroom Teacher Efficacy Toward Implementation of Physical Activity in the D-SHINES Intervention.
Barcelona JM, Centeio EE, Hijazi K, Pedder C.J Sch Health. 2022 Jun;92(6):619-628. doi: 10.1111/josh.13163. Epub 2022 Mar 18.
Lines 106-111. The aims are confused. Has the Physical Literacy Breaks (PLBreaks) program (specifica aim of the study) already been developed?
Materials and Methods
Intervention. Lines 153-168. As for the simple summary and abstract, this part need to be better clarifyed. It is the key part of the study and it is too general.
See, as example:
Masini A, Lanari M, Marini S, Tessari A, Toselli S, Stagni R, Bisi MC, Bragonzoni L, Gori D, Sansavini A, Ceciliani A, Dallolio L. A Multiple Targeted Research Protocol for a Quasi-Experimental Trial in Primary School Children Based on an Active Break Intervention: The Imola Active Breaks (I-MOVE) Study.Int J Environ Res Public Health. 2020 Aug 23;17(17):6123. doi:10.3390/ijerph17176123.
In my opinion a pilot and feasibility study is necessary to value the exercises of the protocol.
Measures. Lines 184-188. Which procedure was followed? Was the Frankfurt plane considered?
Author Response
Dear Reviewer,
Thank you for your review of our manuscript. We have carefully considered your comments and believe that the quality of the paper has improved after incorporating your suggestions. Below are our responses to your suggestions:
Even if the topic is interesting and could provide some new information in the literature the appropriateness of the study design and the description of the intervention are very deficient. The protocol, which is the key part of the study, is too general.
After a careful consideration, I believe that in this form the manuscript is not suitable for publication in this journal.
Simple summary. Lines 27-29 are very confused.
- Author’s response: We have modified those sentences, as suggested. If you feel they are still not understood, please let us know.
Abstract: line 39 (2) Aims: please delete ‘Therefore’
- Author’s response: Deleted, as suggested.
Lines 45-46. As for the simple summary, this part is confused and need to be better clarified: How motivation will be fostered? and how knowledge about healthy lifestyle habits and PA concepts and its benefits will be acquired?
- Author’s response: We understand your comment. We have tried to clarify this in the simple summary and abstract.
Introduction
Line 61. More recent references are needed. See for example:
Worldwide trends in body-mass index, underweight, overweight, and obesity from 1975 to 2016: a pooled analysis of 2416 population-based measurement studies in 128·9 million children, adolescents, and adults.
NCD Risk Factor Collaboration (NCD-RisC).Lancet. 2017 Dec 16;390(10113):2627-2642. doi: 10.1016/S0140-6736(17)32129-3. Epub 2017 Oct 10.
Heterogeneous contributions of change in population distribution of body mass index to change in obesity and underweight. NCD Risk Factor Collaboration (NCD-RisC).Elife. 2021 Mar 9;10:e60060. doi: 10.7554/eLife.60060.
- Author’s response: Added, as suggested.
Line 100. One of the references is quite old. More recent references are needed.
See for example:
Understanding the Benefits of Brief Classroom-Based Physical Activity Interventions on Primary School-Aged Children's Enjoyment and Subjective Wellbeing: A Systematic Review.
Papadopoulos N, Mantilla A, Bussey K, Emonson C, Olive L, McGillivray J, Pesce C, Lewis S, Rinehart N.J Sch Health. 2022 May 23. doi: 10.1111/josh.13196. Online ahead of print.
Classroom Teacher Efficacy Toward Implementation of Physical Activity in the D-SHINES Intervention.
Barcelona JM, Centeio EE, Hijazi K, Pedder C.J Sch Health. 2022 Jun;92(6):619-628. doi: 10.1111/josh.13163. Epub 2022 Mar 18.
- Author’s response: Modified and added, as suggested.
Lines 106-111. The aims are confused. Has the Physical Literacy Breaks (PLBreaks) program (specifica aim of the study) already been developed?
- Author’s response: Thank you for your comment. The PLBreaks program has not been already conducted. Due to this confussion and your suggestion we have rewritten the objective to increase the clarity.
Materials and Methods
Intervention. Lines 153-168. As for the simple summary and abstract, this part need to be better clarifyed. It is the key part of the study and it is too general.
See, as example:
Masini A, Lanari M, Marini S, Tessari A, Toselli S, Stagni R, Bisi MC, Bragonzoni L, Gori D, Sansavini A, Ceciliani A, Dallolio L. A Multiple Targeted Research Protocol for a Quasi-Experimental Trial in Primary School Children Based on an Active Break Intervention: The Imola Active Breaks (I-MOVE) Study.Int J Environ Res Public Health. 2020 Aug 23;17(17):6123. doi:10.3390/ijerph17176123.
- Author’s response: Thank you for your comment. Following your suggestion and the interesting reference provided, we have included futher details about the interventions.
In my opinion a pilot and feasibility study is necessary to value the exercises of the protocol.
- Author’s response: Thank you for your comment. We have included further information regarding the intervention program. In this retard, we have specified the activities to be carried out during the intervention. For the election of the activities we have based on previous studies which found benefitial results. Thus, we truly believe that a pilot study or a feasibility study would not be needed.
Furthermore, transparency and openness, and reproducibility are readily recognized as vital features of science (McNutt, 2014). Nowadays, a substantial proportion of the literature may therefore consist of false or misleading evidence (Ioannidis, 2005; Open Science Collaboration, 2015; Simmons, Nelson, & Simonsohn, 2011). For example, in a transparent science, both null results and statistically significant results are made available and help others more accurately assess the evidence base for a phenomenon. In the present culture, however, null results are published less frequently than statistically significant results (Franco, Malhotra, & Simonovits, 2014). Thus, we decided that the study protocol would be the first step before reporting results of our research.
Nevertheless, we understand your position and we have added this question (the absence of empirical data at this point) as a limitation of the study.
“The present study does not report preliminary data, this study only reports the study protocol that will be performed in the future”.
Measures. Lines 184-188. Which procedure was followed? Was the Frankfurt plane considered?
- Author’s response: Added, as suggested.
Round 2
Reviewer 4 Report
The manuscript has considerably improved after revision.
Minor revision:
Line 107: Please correct ‘stidy’ with study
Figure 1 is not clearly legible. Please improve the quality of the image.
Author Response
Dear Reviewer,
Thank you for your review of our manuscript. We have carefully considered your comments. Below are our responses to your suggestions:
The manuscript has considerably improved after revision.
Minor revision:
Line 107: Please correct ‘stidy’ with study
- Author’s response: Modified, as suggested.
Figure 1 is not clearly legible. Please improve the quality of the image.
- Author’s response: Amended, as suggested.